# Discovery and Validation of Lmj_04_BRCT Domain, a Novel Therapeutic Target: Identification of Candidate Drugs for Leishmaniasis

**DOI:** 10.3390/ijms221910493

**Published:** 2021-09-28

**Authors:** José Peña-Guerrero, Celia Fernández-Rubio, Aroia Burguete-Mikeo, Rima El-Dirany, Alfonso T. García-Sosa, Paul Nguewa

**Affiliations:** 1Department of Microbiology and Parasitology, ISTUN Instituto de Salud Tropical, IdiSNA, Instituto de Investigación Sanitaria de Navarra, Universidad de Navarra, E-31008 Pamplona, Spain; jpena.1@alumni.unav.es (J.P.-G.); cfdezrubio@unav.es (C.F.-R.); aburguetem@unav.es (A.B.-M.); reldirany@alumni.unav.es (R.E.-D.); 2Department of Molecular Technology, Institute of Chemistry, University of Tartu, 50411 Tartu, Estonia

**Keywords:** *Leishmania*, BRCT, drug discovery, therapeutic target, homology modeling, virtual screening, molecular dynamics

## Abstract

Since many of the currently available antileishmanial treatments exhibit toxicity, low effectiveness, and resistance, search and validation of new therapeutic targets allowing the development of innovative drugs have become a worldwide priority. This work presents a structure-based drug discovery strategy to validate the Lmj_04_BRCT domain as a novel therapeutic target in *Leishmania* spp. The structure of this domain was explored using homology modeling, virtual screening, and molecular dynamics studies. Candidate compounds were validated in vitro using promastigotes of *Leishmania major*, *L. amazonensis*, and *L. infantum*, as well as primary mouse macrophages infected with *L. major*. The novel inhibitor CPE2 emerged as the most active of a group of compounds against *Leishmania*, being able to significantly reduce the viability of promastigotes. CPE2 was also active against the intracellular forms of the parasites and significantly reduced parasite burden in murine macrophages without exhibiting toxicity in host cells. Furthermore, *L. major* promastigotes treated with CPE2 showed significant lower expression levels of several genes (*α-tubulin*, *Cyclin CYCA*, and *Yip1*) related to proliferation and treatment resistance. Our in silico and in vitro studies suggest that the Lmj_04_BRCT domain and its here disclosed inhibitors are new potential therapeutic options against leishmaniasis.

## 1. Introduction

Leishmaniasis is among the infectious diseases considered by the World Health Organization (WHO) as neglected tropical diseases (NTDs) [1]. These diseases receive an unevenly low economic support considering their effects on global human health [2]. Present therapies exhibit lack of efficiency, toxicity, or parasite resistance (Appendix A).

Taking into consideration how all available treatments are largely hampered with disadvantages that have made impossible the eradication of this disease, the search and validation of newer therapeutic targets has become a worldwide priority. In this sense, one of the most important advancements in the field of leishmaniasis drug development took place in 2005 with the sequencing of the complete genome of *L. major* [3] that enabled the scientific community to scan the parasite genome to search for suitable drug targets. Such drug targets should allow the development of specific inhibitors or compounds able to tackle the disease with minimum toxicity to the host. There is wide interest in the scientific community to discover new therapeutic targets [4,5] such as leishmanial methionine aminopeptidase 1 [6] or heat shock protein 78 [7].

The availability of the *L. major* genome sequence together with phylogenetic analyses carried out previously revealed the presence of several putative BRCT domains within *L. major*. Some of these domains might exhibit relevant implications for the parasite biology [8]. In fact, previous studies from our group characterized the implications of the protein O97209 (UniProt) bearing Lmj_04_BRCT in *Leishmania* biology [8]. This research linked this domain with *L. major* infectivity and treatment resistance, supporting our further selection of Lmj_04_BRCT for pharmacological targeting. Therefore, homology models were constructed and selected for one of those domains.

On the other hand, BRCT domains are conserved protein regions [9] implicated in a variety of important cell processes including DNA damage repair (DDR) and cell-cycle control [10]. In addition, a few of these protein modules are involved in pathologic processes such as breast and ovarian cancers [11]. Consequently, there are large efforts aiming to discover compounds able to inhibit specifically these protein domains.

Dissimilar to enzyme active sites, interfaces involved in protein–protein interactions (PPIs) are often large, poorly defined, and may be composed of several contact points. Even so, it has been reported that for some PPIs, a selected group of amino acids are responsible for the most important contacts [12]. The targeting of these interactions or “hot spots”, as they are also called, is thought to improve the chances of inhibitor identification. Hot spot contacts are also easier to study due to the possibility to design limited-length peptides to mimic one of the interacting partners. Consequently, the processes of BRCT ligand recognition and inhibition have been extensively studied (Appendix A), and the basis for efficient BRCT inhibition, such as the pSXXF motif [12], VLPF hydrophobic cluster [13], and selective drug-like BRCT (BRCA1) inhibitor bractoppin, have been discovered [14].

The mentioned studies were focused on the design of inhibitors against double-BRCT modules, as it is the case of the BRCA1 protein, but when dealing with single BRCT, there are also reports of small-molecule PPI-inhibitors. (±)-gossypol (especially in its [–] form) exerts its effects as a cell-permeable, small-molecule PPI inhibitor of PARP1. Most likely, gossypol causes dimerization of the protein by covalently binding imines between its dialdehyde and the residues in the PARP1-BRCT domain, inhibiting PARP1 enzymatic activity [15]. Moreover, monoacetylcurcumin is able to specifically bind to the C-terminal BRCT domain of human DNA polymerase λ, which is involved in the repair of DNA double-strand breaks [16].

In this work, we focused on the identification of new compounds with novel chemistry and leishmanicidal activity. To this end, we performed a virtual screening workflow that allowed us to select seven compounds to be tested in vitro. Molecular dynamics (MD) simulations were used to evaluate the stability of the models as well as the interactions with the selected ligands. Pharmacological properties of the selected compounds were predicted and annotated as favorable. After in vitro validation, we were able to identify one active compound among the selected compounds. Leishmanicidal activity was reported for both promastigote and amastigote stages, along with extremely low toxicity in mammalian-derived cells.

These data together with our generated models of BRCT domain allow us to propose a new therapeutic target in *Leishmania*. The results from this work may allow progress in the development of useful drugs against this disease.

## 2. Results

### 2.1. Model Generation and Evaluation

A total of 21 models were generated. Five were generated using web servers, one for each server except that for SwissModel, which was used to generate two different models from two different templates. The others were generated using MODELLER and four different templates along with four different modeling options: (a) no refinement, (b) loop refinement, (c) modeling with ligands, and (d) loop refinement plus modeling with ligands, for a total of 16 more models. After the models were prepared, several known ligands of the BRCT domain were docked as described in the materials and methods. Three models were selected in this way as they docked with better scores to the known ligands than to the decoys. Selected models were also selected for their capability to bind preferably to a higher proportion of known ligands than decoys (highest enrichment factor at 1%). The best results corresponded to the three models: SwissModel_2, Mult1_lig, and Mult1_lr.

The area under the curves (AUC) for the receiver-operator curves (Figure 1D–F) for each model were multi_lr: 0.929; multi_lig: 0.744; and SwissModel: 0.552. These results indicated that the model multi_lr had the best enrichment and distinguished between actives and non-actives.

### 2.2. Sequence Analysis

The selected BRCT sequence was aligned with human and mouse orthologues. Sequence identity was not observed nor expected [17] apart from a few residues. The presence of some elements of secondary structure were also dependent on the selected model. The most variable region was the second alpha helix (α2) which is known to be highly variable and even be missing from some BRCT domains [18,19]. As anticipated, we found high conservation of the five typical hydrophobic BRCT motives: A, B, C, D, and E in the regions already described in the literature [10]; β1 sheet, the loop between α1; and β2, β3, α3 and the C-terminal end, respectively (Figure 2E–G).

Of the three selected models, Mult1_lr and Mult1_lig were the most similar, most likely due to their structural similarity, whereas SwissModel_2 showed the higher differences with the other two. Root mean square deviation (RMSD) (Figure 2A) between Mult1_lr and Mult1_lig was 1.11 Å, between Mult1_lr and SwissModel_2 was 5.70 Å, and between Mult1_lig and SwissModel_2 was 5.80 Å.

The overall structure of the calculated models was compatible with a BRCT domain, considering the highly varied nature that they exhibited. All the models displayed three central β-sheets (Figure 2B–D). The first one (β1) was fairly conserved among the three models and starts from the residues 324–326 to 330–327 of the protein O97209 (UniProt). Afterwards, only the models based on Mult1 alignment (Mult1_lr and Mult1_lig) showed a β-sheet (β2) starting at Arg348 and ending at Ala350. The following β-sheet (β3) was also conserved in all models and started at His360 and ended at Val363. Finally, the last β-sheet (β4) was only present in SwissModel_2 and spanned residues 379–381.

Regarding the α-helixes, the first one (α1) was conserved in the three models and started from Lys336 in SwissModel_2 and Glu332 in the models based on Mult1 (Figure 2B–D). Nevertheless, it ended in Cys345 for all three models. The next α-helix (α2) was only present in the Mult1_lr model and spanned residues 365–367. The next α-helix (α3) was present from residues 383 to 392 in SwissModel_2 and 383–385 in Mult1-based models. Finally, SwissModel_2 showed a last small α-helix comprising residues 398–401.

### 2.3. Molecular Dynamics Simulations

After setup and equilibration in constant volume (NVT) and constant pressure (NPT), no constraints or restraints were imposed on the protein or ligand atoms. Over the course of the simulation, the RMSD of the alpha carbons was calculated against the initial frame. For Mult1_lr the maximum RMSD peaked at 3.84 Å after 46.95 ns, with an average value of 3.02 Å. In the case of Mult1_lig, RMSD peaked at 6.75 Å after 45.8 ns, and its average value was 4.83 Å. Finally, the lowest RMSD values were reported for SwissModel_2 with a maximum of 3.77 Å after 13.09 ns, and an average of 2.84 Å.

Data regarding Mult1_lr as well as the other models: SwissModel_2 and Mult1_lig, are presented in Table 1.

Afterwards, virtual screening was performed, and compounds were selected as presented in Materials and Methods. Compounds CPE1-7 were identified due to their interactions with Mult1_lr (CPE1-4), Mult1_lig (CPE1, CPE5-6), and SwissModel_2 (CPE7).

Upon interaction with the ligands, the maximum and average RMSD increased in both Mult1_lr and SwissModel_2 models. Remarkably, Mult1_lig was the only model whose RMSD was reduced during the course of the simulation upon interaction with the docked ligands, suggesting that the contacts established during the interaction helped to stabilize the protein model (Table 1, Figure 3 and Figure 4). Similarly, the amino acid participation in the protein-ligand interactions was annotated (Figure 5 and Figure 6).

### 2.4. ADME and Bioavailability

Selected compounds display good predicted bioavailability properties:

The theoretical values of log*P* obtained using the SwissADME webserver were within the recommended range. Hence, all the compounds are expected to have a good hydrophilicity/lipophilicity ratio and have potential to be bioavailable compounds. In addition, the compounds were also within every parameter of Lipinski’s rule of five, and the compounds also appeared amenable to further modification given their relatively small size. Data corresponding to these selected compounds appear in following Table 2, Table 3 and Table 4, and Figure 7.

### 2.5. In Vitro Evaluation of Compound Activity in Leishmania major, L. amazonensis, and L. infantum

Finally, from the top-ranked hits from the virtual screening, their actual commercial availability was checked with vendors and seven hit compounds were selected and obtained for experimental testing. First, the antileishmanial activity against the promastigote form of *L. major* of the selected compounds was tested in MTT assays. A wide range of concentrations were tested from 200 µM to 20 nM. Six compounds (CPE1, CPE3-7) did not exhibit leishmanicidal activity at the maximal concentration tested. However, one compound, CPE2, emerged as the most active with an EC_50_ value corresponding to 58.13 ± 1.72 µM. Its chemical structure has an indol-carboxamide group. Since good drug candidates tend to exhibit low toxicity on the host cell, macrophages extracted from murine bone marrow were exposed to growing concentrations of CPE2. Interestingly, this candidate showed no toxicity at the highest tested concentration (200 µM). The selectivity index (SI) of CPE2 was determined as the ratio between the EC_50_ obtained in macrophages and the corresponding EC_50_ in parasites. This value was >3.4 (Table 5). On the other hand, our results showed that CPE2 did not exhibit hemolytic activities (data not shown).In view of the above results, we considered testing the efficacy of CPE2 against the parasite form present in the host, intracellular amastigotes. *Leishmania* infected macrophages were treated at different concentrations corresponding to 4, 20, 40, and 100 µM during 48 h. According to toxicity results, the selected concentrations of the compound showed no toxicity for the peritoneal macrophages. In agreement with the determined CPE2 activity against promastigotes, we detected a reduction in the percentage of infected macrophages compared to that of controls (untreated cells) as well as a significant decrease in the burden of intracellular parasites after treatment with increasing concentrations of the compound (Figure 8).

We also decided to perform a similar experiment in other leishmanial species. The EC_50_ values were 62.44 ± 2.12 and 101.85 ± 0.35 µM in *L. amazonensis* and *L. infantum*, respectively (Table 5).

Our data revealed that the EC_50_ (μM) values for *L. major* and *L. amazonensis* were similar (58.13 ± 1.72 and 62.44 ± 2.12, respectively) and both dissimilar to that of *L. infantum* (101.85 ± 0.35). We were then prompted to perform a phylogenetic analysis focusing on the orthologs of Lmj_04_BRCT in *L. amazonensis* and *L. infantum* (Figure 9). Interestingly, residues Ala335 and His337 in *L.*
*major* (and *L. amazonensis*) are Ser335 and Arg337, respectively, in *L. infantum*. These two highlighted differences in *Leishmania* BRCT sequences seemed to correlate with the aforementioned EC_50_ values of those *Leishmania* species, as Mult1_lr Ser335 showed a high interaction fraction (>0.5) during the MD experiments with docked CPE2 and is likely a key residue in the interaction with this compound. Furthermore, this residue is not conserved among mammals (Figure 9).

### 2.6. Characterization of Mult1_lr–CPE2 Interaction

Considering the promising in vitro activities of compound CPE2, we aimed to further analyze the interaction between Lmj_04_BRCT (Mult1_lr) and compound CPE2. To this end, we explored the RMSF and MD dynamic correlations of the docked structure (Figure 10).

Figure 11A shows the dynamical correlation between all residues in the complex of multi_lr and inhibitor CPE2 during the MD simulation. It can be observed that the initial residues had a positive correlation with residues in domain regions 35–40 and 55–60, in addition to those regions among themselves (35–40 with 55–60). In addition, there were regions of anticorrelation (the distances between residues increased among themselves) for region 20–25 with other regions except with 35–40 and 55–60, as well as with the terminal residues.

The root mean squared fluctuation for each residue (RMSF, Figure 11B) showed that most residues did not change their position much throughout the simulation. Those that most fluctuated were in positions 25, 50, and 65, in addition to the terminal residues. The protein’s secondary structure also showed the α-helix and β-sheet regions. Remarkably, the average RMSD for the residues of BRCT domain that interacted with the compound was 1.00 Å, suggesting a stable protein–ligand complex throughout the simulation, and in particular, in the binding site region.

### 2.7. CPE2 Exposure Significantly Altered the Gene Expression of Leishmania major ABC Transporter, α-Tubulin, CYCA, and Yip1

The effect of CPE2 exposure on *L. major* gene expression was then analyzed. For this purpose, a set of 19 genes belonging to different biological pathways were selected and a *L. major* culture was treated with CPE2 (50 μM) for 48 h. mRNA expression level was significantly decreased for *ABC transporter H1* (Figure 12A). 

Further, we investigated the effects of CPE2 on *Leishmania* genes associated with proliferation. We detected a significant decrease of *α-tubulin* and *CYCA* expression (Figure 12B).

Recently, Patino et al. published a list of dysregulated genes after in vitro generation of antimony resistance [20]. Consequently, we were prompted to analyze the effects CPE2 (50 μM, 48 h) on a set of *Leishmania* genes. Interestingly, we found a significant reduction in the expression of both *CYCA* and *Yip1* genes (Figure 12C).

## 3. Discussion

To find new targets and drugs against leishmaniasis, we aimed to construct Lmj_04_BRCT homology models to screen novel chemical compounds and to test them in vitro. It has been reported that the quality of the homology models increases when incorporating information from interaction with known ligands [21,22]. Furthermore, the docking enrichment of known ligands among the top-scoring molecules from a pool that included a large number of decoys with similar properties is regarded as a criterion of model accuracy [23].

Considering the fact that there are no known inhibitors of Lmj_04_BRCT domain, our strategy was then based on phosphorylated ligands, as phosphate-dependent interactions are one of the most important features of this domain [24]. To this end, we incorporated one phosphorylated peptide [25] and one phospho-mimicking peptide [26] to the set of known ligands. Since Lmj_04_BRCT is a single domain, we therefore included the inhibitor of the single BRCT, gossypol [15], and the interaction of a single BRCT with a phosphorylated peptide [27] to our calculations. Due to its specificity to BRCA1 tandem BRCT domains, bractoppin was used as a negative control in the docking experiments. Notably, our docking results favored the single BRCT inhibitor gossypol among other structures in the known ligands set.

SwissModel_2 seemed to have better quality scores that the other two. However, the three models were used for the in vitro tests due to their good docking results. With regards to the structure, the generated models selected for this study fitted the previously reported BRCT structures, which usually show a hydrophobic core of β-sheets trapped between α-helixes with a typical β1, α1, β2, β3, α2, β4, and α3 pattern [18,28], α2 being variable among BRCT domains [18,29]. The models selected for this study clearly fitted the aforementioned criteria. Interestingly, the Mult1_lr model contained both β2 sheet and the α2 helix, whereas the expected β4 was only present in SwissModel_2. Furthermore, the loop refinement region of model Mult1_lr spawned residues 367–382 which fell between α2 and α3 helixes in both models this region was selected during intermediate steps of modeling with MODELLER and this refinement might have helped the correct folding of this α2 helix which was not present in SwissModel_2. Thus, Mult1_lr seemed to be the best model. Further, we searched for and discovered the hydrophobic motives present in conventional BRCT domains [19], which would support the typical BRCT folding harbored in our models as these hydrophobic interactions are important for correct BRCT folding [18].

As aforementioned, the protein O97209 (UniProt) bearing Lmj_04_BRCT was described to be involved in *L. major* infectivity and treatment resistance [8]. There are multiple examples of targeting individual essential parasite proteins, such as Hsp90 [30], *L. major* Pteridine reductase 1 (LmPTR1) [31], or *Leishmania* arginase in order to kill the parasite minimizing the harm to the host cell [22,32,33]. Therefore, we decided to use our generated models to perform virtual screening and assays to find possible inhibitors of this important domain in *L. major*.

We analyzed the stability of generated BRCT models by recording RMSD, RMSF, and DCCM over the course of MD simulations. Overall, the most stable model was SwissModel_2, followed by Mult1_lr and Mult1_lig. One possible explanation for the low stability of the Mult1_lig model would be that the model was built with an existing ligand that contributed to its stability and the protein was more flexible or disordered without it, as described in other previous studies [34,35]. In addition, our study showed that the average RMSD values corresponding to Lmj_04_BRCT models were similar to those reported for other BRCT domains ranging 3–4 Å [36].

The physicochemical properties of the selected ligands [37] showed excellent predicted bioavailability properties, such as Lipinski’s rule of five, susceptibility to metabolic inactivation, and barrier permeability. Furthermore, these compounds seemed to have easily feasible synthesis and be amenable to further modification. On the other hand, considering the low toxicity and blood–brain barrier permeability of the selected compounds, especially CPE2, it may be interesting to evaluate the activity of these compounds against other trypanosomatids, such as *Trypanosoma cruzi* and *T. brucei,* etiologic agents of Chagas diseases, and sleeping sickness, respectively [38,39]. These remain future directions to be explored.

We then analyzed the MD trajectories of the selected compounds docked in the models and compared the data with those of the isolated models. There was no observed reduction in the average RMSD values of SwissModel_2 and Mult1_lr models after exposure with the ligands. This is probably due to the canonical features of the BRCT domains since their folding and stability do not depend on their interaction with a ligand [18,40]. For example, it is well known that, at least for tandem BRCT domains, the BRCT domains are already folded before the initiation of the binding process. Thus, it only exhibits a moderate re-arrangement of its side chains to allow interactions with the ligand [41]. Furthermore, it has been reported that low RMSD values after docking calculations offer better correlations among in vitro and in silico experiments [42] thus, supporting our docking results.

Furthermore, the reference structure of single interacting BRCT domain [27] was aligned using PyMOL [43] with our constructed Lmj_04_BRCT models. Considering amino acid position and properties, we annotated Ser330, Arg331, Tyr353, Val354, Pro355, Lys374, Leu378, and Tyr380 as the leishmanial equivalents for key mediators of Lmj_04_BRCT interaction. By considering the interaction fraction, some of the predicted amino acids for involvement in the interaction with the ligand were validated. Six (Ser330, Tyr353, Pro355, Lys374, Leu378, and Tyr380) of those eight residues exhibited high interaction fractions values. Such a phenomenon was especially relevant for residues Ser330 and Tyr353, as they were reported to be relevant for the interaction process in the models Mult1_lr and SwissModel, whose docking study was performed without constrains and both residues were present in both models. Furthermore, the interaction with Lys374 and Tyr380 was reported for both Mult1_lig and SwissModel models in the presence of compounds.

Afterwards, considering that all the selected compounds were predicted to exhibit favorable physicochemical and development properties, we evaluated the antileishmanial potential of the selected docked compounds. To our knowledge and after searching abstracting databases, this is the first time that the presented compounds were proposed as potential leishmanicidal agents, nor have they been described elsewhere. Data collected proved that among those tested, the novel compound CPE2 showed a reasonably potent inhibition against *L. major* and *L. amazonensis* and was active against *L. infantum*, although with a lower effect compared to the reference drug Amphotericin B [44,45], which despite of its excellent antileishmanial properties is not free of drawbacks [46]. Nevertheless, these novel compounds most likely will need additional pharmacological modifications to enhance their leishmanicidal activity. Interestingly, this is predicted to be feasible by the predicted synthetic accessibility of compound CPE2, as well as possibilities for decoration on the rings and further isosteric replacements

With regards to the difference in activity between *L. major, L. amazonensis*, and *L. infantum,* we annotated that the top performing compound, CPE2, was less active in *L. infantum* than in *L. major*. In fact, the EC_50_ value increased almost twice. The sequence homology analysis showed that *L. major* and *L. infantum* orthologs share a very high identity of BRCT sequence. However, the sequence alignment revealed that Ala335 and His337 in *L. major* were substituted by a serine and an arginine, respectively, in *L. infantum*, whereas these same residues were conserved in *L. amazonensis* exhibiting a similar sensitivity. Our data suggested that some selected amino acids might be important for the interaction between Lmj_04_BRCT and its ligands.

In murine bone marrow-derived macrophages (BMDMs), CPE2 exhibited low toxicity, and considering its wide SI, we were prompted to test CPE2 against the intracellular forms of the parasite.

*L. major* and murine-derived macrophages were used in in vitro infection experiments and found that parasite load was significantly diminished at concentrations even lower than promastigote EC_50_, which confirmed the activity of CPE2 in both promastigote and amastigote forms of the parasite. Interestingly, the activity on amastigotes reached its maximum at 40 µM or even lower, as there is no increase in activity after the treatment with concentration > 100 µM. The activity on amastigote forms at concentrations lower than promastigote EC_50_ suggested that CPE2 might exert its leishmanicidal effects through secondary routes other than the direct elimination of the parasite. One possible option would be the modulation of the immune response in the macrophage, which has already been reported for a several well-known existing antileishmanial compounds, such as antimony-based compounds [47,48,49] and miltefosine [50,51,52,53,54]. Another option may be the metabolization of the compound into a more active chemical species, as previously described with antimony derivatives, where there is evidence of the *in vivo* chemical reduction of Sb(V) into the more active Sb(III) [55,56,57], most likely by thiols [58,59,60].

The novel leishmanicidal compound CPE2 showed activity on both intracellular and extracellular stages of the parasite, and against several *Leishmania* species. When dealing with the development of CPE2 as a novel antileishmanial drug, further tests need to be performed before the realization of clinical trials with this compound. These assays include the optimization of CPE2, as well as preclinical studies using *in vivo Leishmania* infections. By using experimental models in drug development, pharmacokinetics, pharmacodynamics (PK/PD), safety properties and other key candidate drug parameters would be assessed [61]. The predicted in silico binding of CPE2 and Lmj_04_BRCT reinforced the idea that the biological basis for this antileishmanial effect may indeed be Lmj_04_BRCT inhibition. Therefore, the generated structure for Lmj_04_BRCT could also be used for the development of more lead drugs candidates to fight this disease.

Finally, drug combination has been proposed as an alternative for leishmaniasis treatment [61,62]. A synergistic effect may allow the administration of drugs below their original dose limits. Considering the known toxicity of the current treatments, this possibility encompasses potential advantages such as a higher efficacy, shorter treatment regimens, higher drug tolerance, and less side effects [61,62]. Furthermore, drug combination has also been evaluated as a tool for the reduction of resistances [63]. Considering the aforementioned data, it seems promising to test the activity of the selected compound CPE2 in combination with other antileishmanial compounds.

## 4. Materials and Methods

### 4.1. Structure Prediction and Evaluation

Leishmanial BRCT protein sequences were retrieved from UniProt database [64]. Among the 12 *L. major* BRCT-bearing proteins annotated in UniProt (UniProt IDs: O97209, E9AE40, Q4Q971, Q4Q1J9, Q4Q8S5, Q4Q9T2, Q4QHI6, E9AE58, Q4Q2B7, Q4QAZ7, Q4QIW2, and E9AFV6) (search performed in March 2021), we selected O97209 for being the only protein which had been manually reviewed by UniProtKB curators [64]. Since O97209 is encoded by a gene located in chromosome 04, we renamed our target sequence as Lmj_04_BRCT.

Lmj_04_BRCT domain boundaries within O97209 were taken from the Conserved Domains Database (CDD) (CDD: 406888) [65]. Protein sequences belonging to Lmj_04_BRCT were selected and homology modeling was performed using MODELLER9.20 [66,67,68,69] software as well as Itasser [70,71], Robetta [72,73], Modweb [74,75], and SwissModel [76,77,78] web servers. Two different templates for SwissModel were selected according to the highest available protein X-ray crystal structure resolution and highest available sequence homology. Modeling using web servers was performed using default options, whereas scripts for MODELLER were downloaded from the program’s webpage and modified to suit our input sequences (https://salilab.org/modeller/, accessed on 15 September 2018).

All generated models were prepared using the default options of Protein Preparation Wizard [79] available in Maestro2018 (www.schrodinger.com, accessed on 1 January 2019), except for the following exceptions: Hydrogens were added after removal of any original positions, zero-order bonds to metals were not generated as they were not present within the models, selenomethionines were converted to methionines, and sidechains were filled-in using Prime; heteroatomic states were generated using Epik at pH 7 ± 2. Hydrogen bonds were interactively optimized for all models. Lastly, a restrained minimization (RMSD of heavy atoms: 0.3 Å) was performed using the OPLS3e force field.

The binding site (docking box) was defined as the space extending 25 Å from amino acids participating in the single-BRCT domain interaction (Ser 657, Gln 658, Phe 681, Phe 682, Met 692, Lys 707, Ala 710, and Trp 714 of human Topoisomerase IIβ binding protein 1 [TopBP1] [PDB ID: 5U6K]) [27]. Known ligands were also identified from the literature (Table 6) [15,25,26]. Decoys for small molecule drugs were generated using the DUD-E database [80,81], whereas peptide decoys were generated by inversion and alanine substitution of individual amino acids. Ligand preparation was performed with LigPrep [79] with the following modifications: Original state was included after ionization and chiralities were determined from 3D structure. Docking was performed with Glide XP (extra precision) [82,83,84], including input ring conformations. Epik state penalties were not added to the docking score. Models were selected by their capability to bind to known ligands rather than decoys as measured by their enrichment factors. Specific details regarding the selected models were presented in Table 7.

### 4.2. Virtual Screening Analysis

For virtual screening, a previously prepared chemical library was used consisting of a set of 266,435 commercially available compounds that had been filtered for toxic or reactive groups, such as epoxides, hydrazines, etc., as well as those molecules with undesirable predicted solubility, high number of aromatic rings, and high number of rotatable bonds [85]. This resulted in fully prepared compounds with hydrogens assigned in an ionic state corresponding to a pH of 7 [86,87,88]. Virtual screening was also performed with Glide XP, similarly to the docking experiments, and Epik state penalties were not used. The top 0.01% scoring interactions (*n* = 300 compounds) were manually inspected and filtered according to the following criteria: Presence among the other top scoring models (the same compound is highly ranked among several homology models), the presence of at least two non-hydrophobic protein–ligand interactions, and noncovalent interactions through π electron-rich groups, as they provide significant stability and selectivity to the compounds [34,89].

### 4.3. Molecular Dynamics

The OPLS_2005 force field for the protein models was employed. The predicted structure of BRCT was solvated in an orthorhombic box of water. The distance between the structure and the edge of the box was set to 10 Å and the box volume was minimized. The net charge of the protein was automatically neutralized by adding counter-ions to reach a concentration of 0.15 M NaCl salt. MD simulations had a minimum duration of 50 ns and were carried out using DESMOND [90,91]. Atom positions were written to file every 1200 ps. To follow the simulations, the RMSDs of the Cα atoms with respect to the minimized starting structure were calculated and monitored. Data analysis for the MD simulations was carried out using the Simulation Interaction Diagram tool implemented in the DESMOND MD package [90,91], and the simulation trajectory was also analyzed with the package Bio3D [92] for calculating root means square fluctuations of residues (RMSF), and dynamical cross-correlation matrix from Cartesian Coordinates (DCCM) plots.

### 4.4. In Vitro Biological Evaluation

#### 4.4.1. Compounds

Experimentally studied chemicals were purchased from MolPort, Inc. (Riga, Latvia). Powder compounds were dissolved in dimethyl sulfoxide (DMSO) at a concentration of 0.001 M. The sterile filtration was achieved using 0.2-µm filter disks. Serial dilutions with supplemented medium were prepared daily to a final concentration of less than 2% DMSO in cell culture.

#### 4.4.2. Cells and Culture Conditions

*Leishmania major* promastigotes (Lv39c5) were kindly provided by Manuel Soto (Centro de Biología Molecular Severo Ochoa (CSIC-UAM), Madrid, Spain) and were grown at 26 °C in M199 medium supplemented with 25 mM of HEPES (pH 7.4), 0.1 mM of adenine, 0.0005% (*w*/*v*) hemin, 1 mg/mL of biopterin, 0.0001% (*w*/*v*) biotin, 10% (*v*/*v*) heat-inactivated fetal calf serum (FCS), and an antibiotic cocktail (50 U/mL of penicillin, 50 mg/mL of streptomycin). To maintain their infectivity, *Leishmania* parasites were isolated from infected female BALB/c mouse spleen and maintained in culture for not more than five passages.

Murine bone marrow-derived macrophages (BMDMs) were obtained from female BALB/c mice as previously described [93]. Briefly, BMDMs were generated from bone marrow stem cells obtained by flushing femur and tibia with phosphate buffered saline (PBS) and cultured on 10-cm-diameter “bacteriological” plastic plates for seven days in Dulbecco’s modified Eagle medium (DMEM) supplemented with 10% (*v*/*v*) heat-inactivated FCS, penicillin/streptomycin, and 20% filtered supernatant from the L929 cell line as a source of granulocyte/macrophage colony stimulating factor. All the procedures involving animals were approved by the Animal Care Ethics Commission of the University of Navarra (approval number: E5-16(068-15E1) 25 February 2016).

#### 4.4.3. Leishmanicidal Activity

##### Activity against Promastigotes

To determine the antileishmanial activities of the compounds analyzed in this study, exponentially growing cells (2 × 10^6^
*L. major* promastigotes/mL) were seeded in 96-well plates (100 μL per well) with increasing concentrations of the compounds diluted in 100 µL of M199 and maintained at 26 °C. After 48 h of incubation, the decrease in the number of viable cells was determined using the colorimetric assay with 3-[4,5-dimethylthiazol-2-yl]-2,5-diphenyltetrazolium bromide (MTT) (Sigma, St. Louis, MO, USA) [94]. MTT solutions were prepared at 5 mg/mL in PBS, filtered, and maintained at −20 °C until use. After adding 100 μg/well of MTT, plates were incubated for 4 h under the same conditions. Then, 80 μL of DMSO was added to each well and plates were slightly shaken to dissolve formazan crystals. The optical density (OD) was measured in a Multiskan EX microplate photometer plate reader at 540 nm and the half-maximal effective concentration (EC_50_) was calculated. The EC_50_ represents the concentration that gives half-maximal viability of treated cells with respect to untreated cells (controls). This parameter was obtained by fitting a sigmoidal E^max^ model to dose–response curves.

##### Activity against Intracellular Amastigotes

Differentiated BMDMs were removed from the substrate by pipetting of ice-cold PBS, seeded in 8-well culture chamber slides (Lab-Tek; BD Biosciences) at a density of 5 × 10^4^ cells per well in DMEM medium, and allowed to adhere overnight at 37 °C in a 5% CO_2_ incubator. In order to perform the infection assay, *L. infantum* and *L. amazonensis* promastigotes as well as metacyclic *L. major* promastigotes isolated by the peanut agglutinin method [95] were used to generate infection at a macrophage/parasite ratio of 1/20. The plates were incubated for 24 h under the same conditions to promote promastigote phagocytosis. The wells were then washed with medium to remove the extracellular promastigotes and plates were incubated with fresh medium supplemented with increasing concentrations of compounds. After 48 h, cells were washed with PBS, fixed with ice-cold methanol for 5 min, and stained with Giemsa stain. To determine parasite burden, the number of amastigotes per infected macrophages was counted under a light microscope. The mean number of amastigotes per infected macrophage was determined by dividing the total number of amastigotes counted by the number of infected macrophages.

#### 4.4.4. In Vitro Toxicity Evaluation and Hemocompatibility

To determine the cytotoxicity of selected compounds, BMDMs obtained as explained before were used for the study and an MTT test was performed as described above. Briefly, 5 × 10^4^ cells were seeded per well in 96-well plates and allowed to adhere for 24 h at 37 °C in a 5% CO_2_ humidified atmosphere. The culture medium (DMEM) was replaced by fresh medium with increasing concentrations of compounds and, after 48 h of incubation; the EC_50_ was calculated as described above.

The potential hemolytic activity of CPE2 was evaluated following a protocol previously described [96]. Human erythrocytes from healthy volunteers were concentrated by centrifugation (10 min at 250× *g*, 4 °C), and the resulting pellet was diluted 1:5 in PBS. Erythrocytes were exposed to increasing concentrations of CPE2 (0, 0.78, 1.56, 3.12, 6.25, 12.5, 25, 50, 100, and 200 µM) and incubated for 30 min at 37 °C, with shaking at 100 rpm. Lastly, erythrocyte solutions were centrifuged at 250× *g* for 10 min and the amount of hemoglobin present in the supernatant was quantified by spectrometry at 540 nm. Negative and positive controls of hemolysis were prepared by suspending the erythrocyte solution in PBS or 10% Triton, respectively.

### 4.5. Statistical Analysis

Statistical analysis was performed using PRISM version 5.0 (GraphPad Software Inc., San Diego, CA, USA). The data are presented as means ± SD. Comparisons between two groups were made using the two-tailed, unpaired t-test. Statistical significance was determined (***, *p*  <  0.001; **, *p*  <  0.01; *, *p*  <  0.05).

### 4.6. Calculation of Selectivity Index

Three different assays were performed to calculate the selectivity index (SI) of selected compounds, which was determined as the ratio between the EC_50_ obtained in macrophages and the corresponding EC_50_ in parasites.

### 4.7. BRCT Sequence Identification, Orthologue Annotation, and Alignments

Lmj_04_BRCT amino acid sequence was downloaded from UniProt database (ID: O97209) [64]. BRCT domain boundaries were taken from the Conserved Domains Database (CDD) (CDD:293197) [65].

Orthologue BRCT domain sequences were found using the tblastn algorithm in the Basic Local Alignment Search Tool (BLAST) (BLAST [Internet]. Bethesda (MD): National Library of Medicine (US), National Center for Biotechnology Information; 2004. Available from: https://blast.ncbi.nlm.nih.gov/Blast.cgi accessed on 1 August 2021) program using the aforementioned *L. major* BRCT protein sequence as query. Multiple sequence alignments were performed using MUSCLE v3.8.31 [97,98], as implemented in the European Molecular Biology Laboratory-European Bioinformatics (EMBL-EBI) web server [99].

### 4.8. Gene Expression Quantification

RNA from treated and control cells was extracted using TRIZOL reagent [85,100]. Leftover DNA was removed using Ambion DNA-free Kit (Invitrogen, Vilnius, Lithuania) following manufacturer’s instructions. Reverse transcription was performed with 1 µg of RNA using SUPERscript II Reverse Transcriptase (18064-014 Invitrogen) following the protocol of the manufacturer. The obtained cDNA was then used for mRNA quantification.

qPCR assays were performed in 96-well plates using a 7500 real-time PCR system (Applied Biosystems, Foster City, CA, USA), (Applied Biosystems) with SYBR Green PCR master mix (Applied Biosystems) according to manufacturer’s instructions. Specific primers for selected genes were designed using Primer-BLAST [101]. The complete set of primers is available in Appendix A. Fluorescence readings were performed at the elongation temperature (60 °C). *Glyceraldehyde-3-phosphate dehydrogenase* (*GAPDH*) (Appendix A) was used as the housekeeping gene to normalize *L. major* gene expression. The amount of each transcript was expressed by the formula 2*ct*(GAPDH) − ct(gene), with *ct* indicating the point (PCR cycle) at which the fluorescence rises appreciably above the background fluorescence [102]. The gene expression levels of the target genes under CPE2 treatment were compared to those obtained during no treatment conditions.

## Figures and Tables

**Figure 1 ijms-22-10493-f001:**
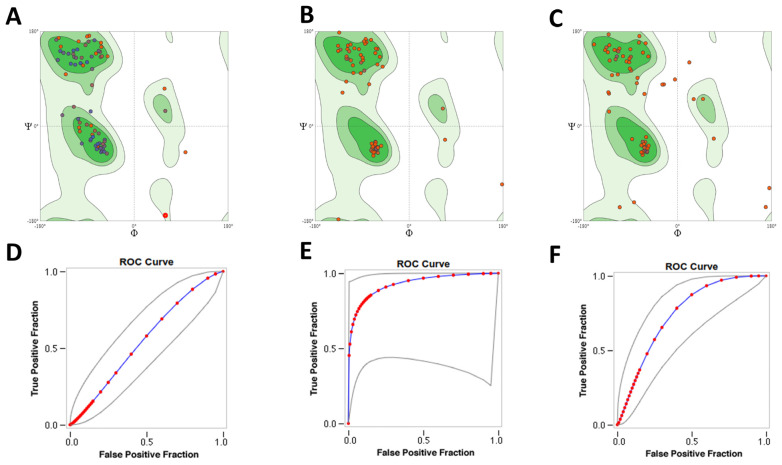
Evaluation of selected Lmj_04_BRCT models. Ramachandran plots for: (**A**) SwissModel_2; (**B**) Mult1_lr; (**C**) Mult1_lig. SwissModel_2 and Mult1_lr contained 92.31% of its amino acids in the favored regions, whereas Mult1_lig contained 75.00% of its amino acids in the favored regions. Receiver-operator curves for docking to models: (**D**) SwissModel_2, (**E**) Mult1_lr, and (**F**) Mult1_lig.

**Figure 2 ijms-22-10493-f002:**
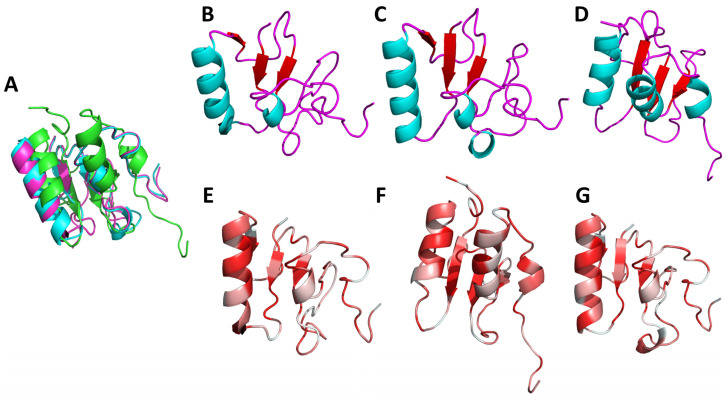
Selected Lmj_04_BRCT models secondary structure and hydrophobic regions. Ribbon representation of (**A**) aligned models (green: SwissModel_2, pink: Mult1_lig, cyan Mult1_lr); (**B**) Mult1_lig; (**C**) Mult1_lr; (**D**) SwissModel_2. Colors represented secondary structure elements: Red: β-sheets. Cyan: α-helixes. Pink: loops. Usual BRCT hydrophobic core of β-sheets trapped between α-helixes in a β1, α1, β2, β3, α2, β4, and α3 disposition is only moderately represented in the constructed models as only the Mult1_lr model folded into these expected β2 sheet and α2 helix whereas expected β4 was only present in SwissModel_2. Ribbon representation of (**E**) Mult1_lig; (**F**) Mult1_lr; (**G**) SwissModel_2. Higher intensity of red color represented higher hydrophobicity. Conventional BRCT hydrophobic clusters (β1, α1 and α1-β2 loop, β3 sheet, α3 helix, and C-terminal region) were also found in the constructed models.

**Figure 3 ijms-22-10493-f003:**
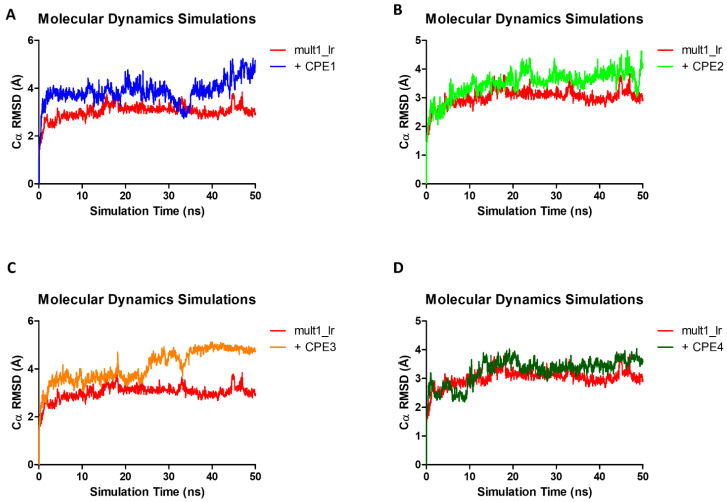
Molecular dynamics analysis of the stability of Mult1_lr model when isolated and when interacting with different selected ligands. (**A**) CPE1, (**B**) CPE2, (**C**) CPE3, (**D**) CPE4. Red: isolated Mult1_lr BRCT model, blue: Mult1_lr model bound to CPE1, light green: Mult1_lr model bound to CPE2, orange: Mult1_lr model bound to CPE3, dark green: Mult1_lr model bound to CPE4. Presented data were α-Carbons RMSD over 50 ns of MD simulation.

**Figure 4 ijms-22-10493-f004:**
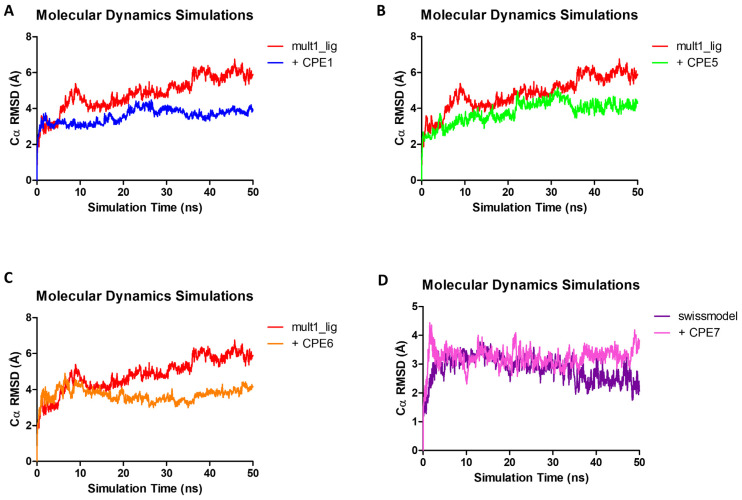
Molecular dynamics analysis of the stability of Mult1_lig and SwissModel_2 model when isolated and when interacting with different selected ligands. (**A**) CPE1, (**B**) CPE5, (**C**) CPE6, (**D**) CPE7. Red: isolated Mult1_lig BRCT model, blue: Mult1_lig model bound to CPE1, light green: Mult1_lig model bound to CPE5, orange: Mult1_lig model bound to CPE6, purple: isolated SwissModel_2 BRCT model, pink: SwissModel_2 model bound to CPE7. Presented data were α-Carbons RMSD over 50 ns of MD simulation.

**Figure 5 ijms-22-10493-f005:**
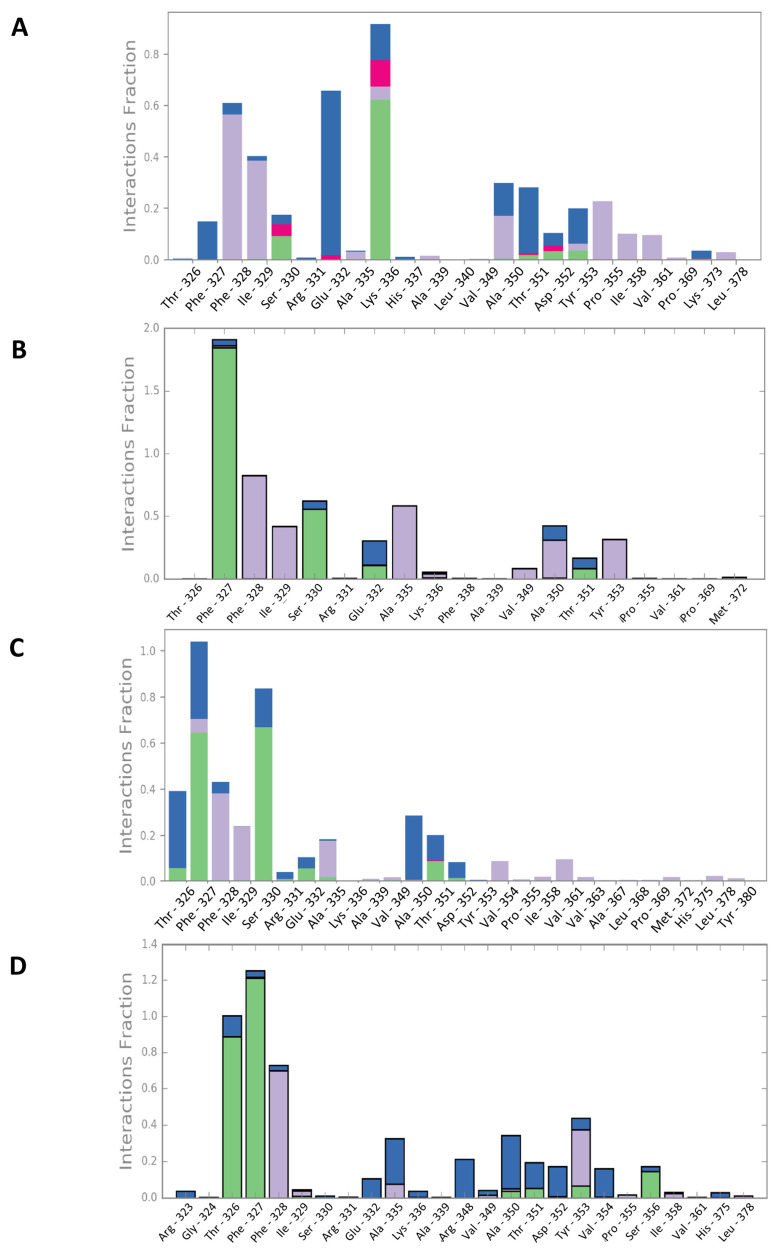
Persistence of the model–ligand interactions found within the constructed Lmj_04_BRCT models and selected compounds during the 50 ns molecular dynamics simulation. (**A**) CPE1-Mult1_lr, (**B**) CPE2-Mult1_lr, (**C**) CPE3-Mult1_lr, (**D**) CPE4-Mult1_lr. Green: hydrogen bonds. Purple: hydrophobic interactions. Red: ionic interactions. Blue: water bridges.

**Figure 6 ijms-22-10493-f006:**
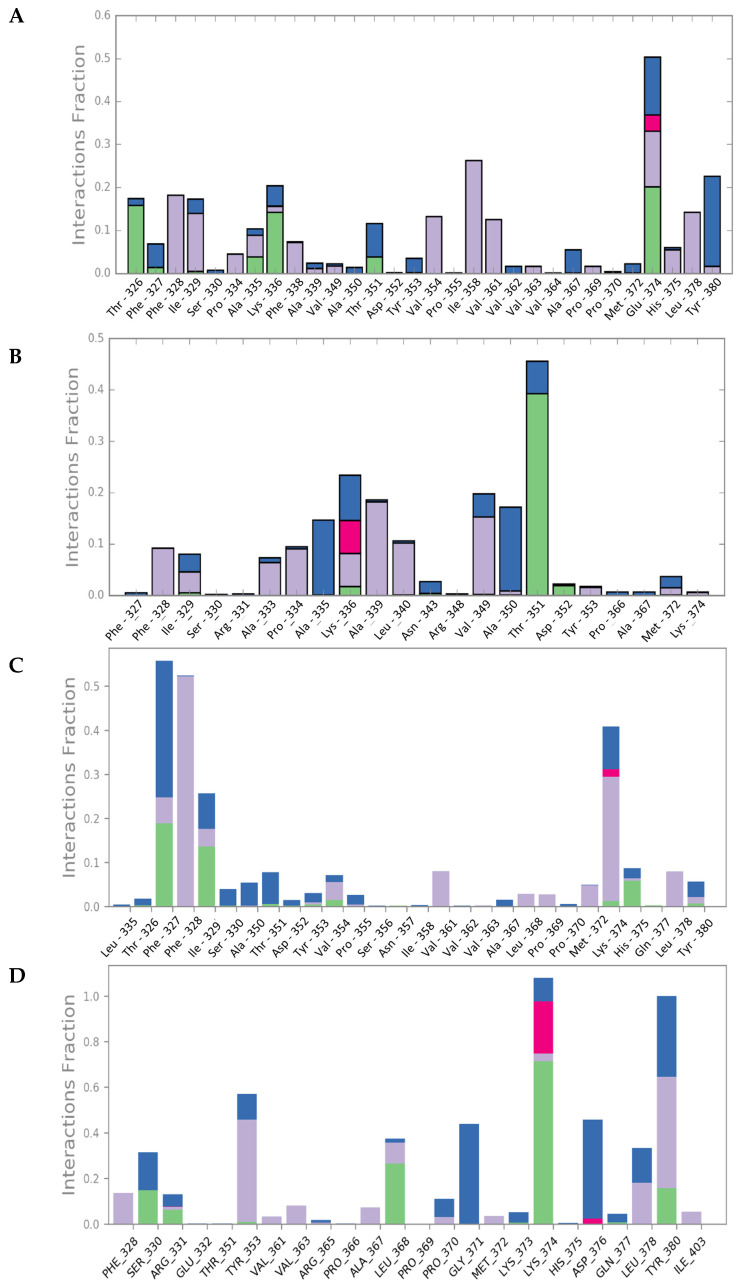
Persistence of the model–ligand interactions found within the constructed Lmj_04_BRCT models and selected compounds during the 50 ns molecular dynamics simulation. (**A**) CPE1-Mult1_lig, (**B**) CPE5-Mult1_lig, (**C**) CPE6-Mult1_lig, (**D**) CPE7-SwissModel_2. Green: hydrogen bonds. Purple: hydrophobic interactions. Red: ionic interactions. Blue: water bridges.

**Figure 7 ijms-22-10493-f007:**
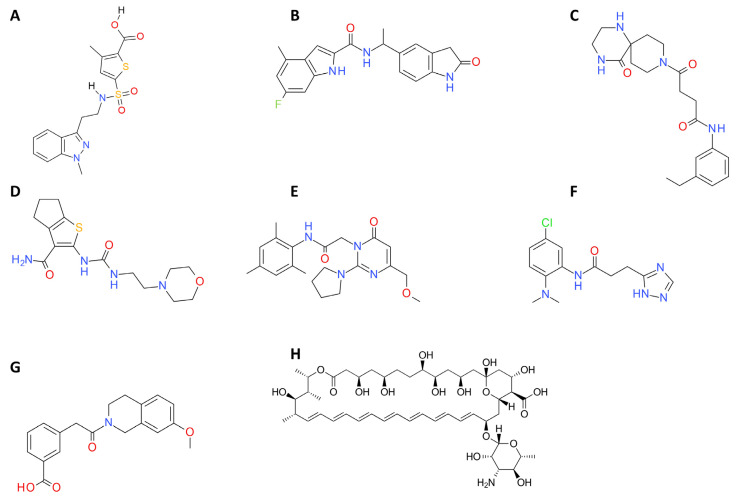
Structures for the in silico identified compounds selected for further in vitro testing. (**A**) CPE1, (**B**) CPE2, (**C**) CPE3, (**D**) CPE4, (**E**) CPE5, (**F**) CPE6, (**G**) CPE7, (**H**) amphotericin B.

**Figure 8 ijms-22-10493-f008:**
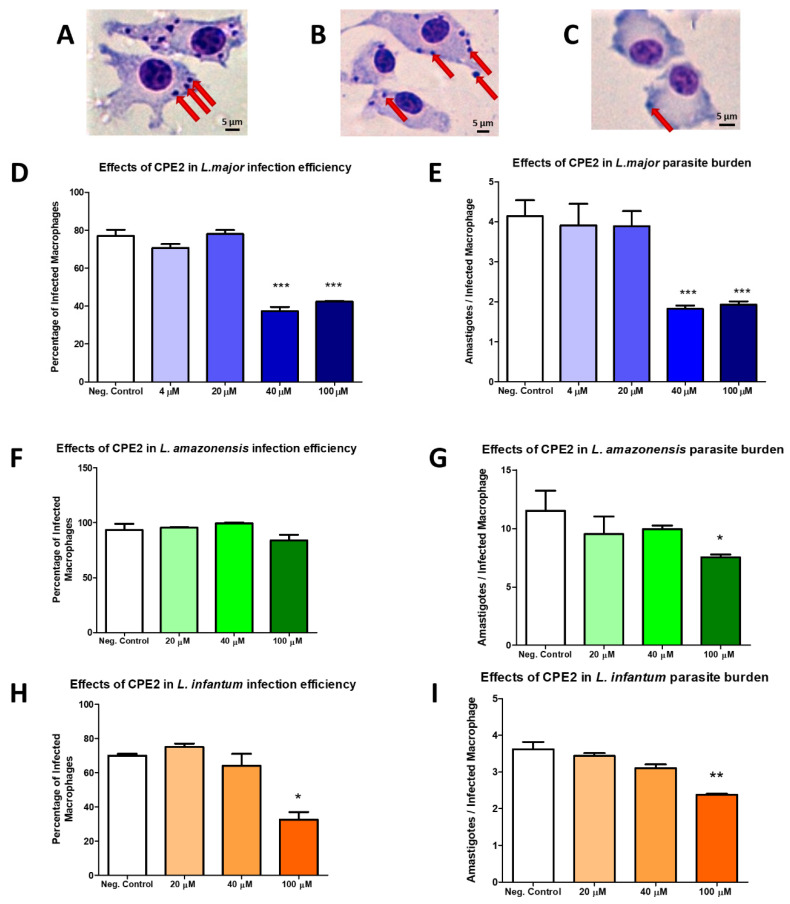
Effects of CPE2 on intracellular amastigotes of *L. major, L. amazonensis*, and *L. infantum*. Representative pictures of (**A**) *L. major* in vitro infection negative control, (**B**) *L. major* in vitro infection treated with 20 μM CPE2, (**C**) *L. major* in vitro infection treated with 100 μM of CPE2. The red arrows indicated the amastigotes. The scale bar in black represented 5 µm. The drug treatment was performed at four different concentrations: 4, 20, 40, and 100 μM CPE2. Bars represented the percentage of infected macrophages (**D**,**F**,**H**) or the number of parasites per infected cell (**E**,**G**,**I**) and after 48 h of treatment. Negative controls represented the untreated cells. The results showed the means from three independent duplicate experiments ± SD. Significant reductions in the proportion of infected amastigotes, as well as in the count of amastigotes per infected macrophage were annotated (*, *p*  <  0.05. **, *p*  <  0.01. ***, *p*  <  0.001).

**Figure 9 ijms-22-10493-f009:**
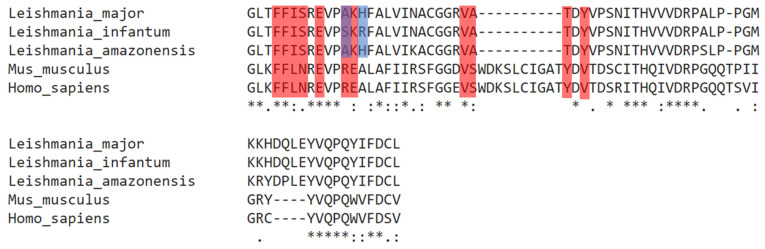
Multiple sequence alignments carried for *Leishmania major* (Lmj_04_BRCT) sequence and orthologues in *L. infantum, L. amazonensis, Mus musculus*, and *Homo sapiens*. Red: Important residues for CPE2–Mult1_lr interaction. Blue: Different residues between Lmj_04_BRCT domain protein sequences from *L. major* and its orthologue from *L. infantum*. An asterisk (*) represents positions which have a single, fully conserved residue. A colon (:) represents conservation between groups of highly similar properties. In contrast, a full stop (.) represents conservation between groups of weakly similar properties.

**Figure 10 ijms-22-10493-f010:**
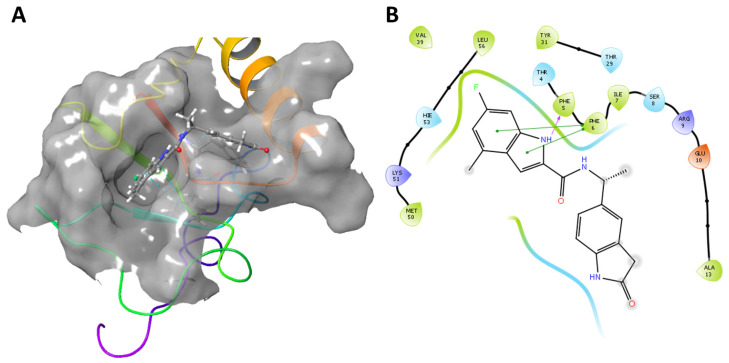
3D (**A**) and schematic 2D (**B**) representations of the interactions between selected top-scoring compound CPE2 and Mult1_lr model. (**A**) Yellow: hydrogen bonds, blue: Pi-stacking. (**B**) Purple: hydrogen bonds, green: Pi-stacking.

**Figure 11 ijms-22-10493-f011:**
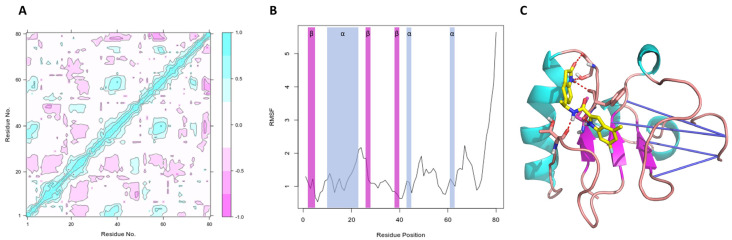
Analysis of Mult1_lr-CPE2 MD experiments. (**A**) Correlation matrix for all residues during the molecular dynamics simulation trajectory for multi_lr in complex with CPE2. (**B**) Root mean square fluctuation (RMSF, Å) for all the residues in the MD simulation of multi_lr in complex with CPE2. Blue-shaded regions corresponded to alpha helices, magenta-shaded regions corresponded to beta sheets. (**C**). Dynamic cross-correlation matrix (DCCM) vectors acting on the MD simulation of the structure of the complex of multi_lr and inhibitor CPE2 (in yellow). Hydrogen bonds are shown in red, alpha helices in cyan, beta sheets in magenta, and anticorrelation vectors in indigo.

**Figure 12 ijms-22-10493-f012:**
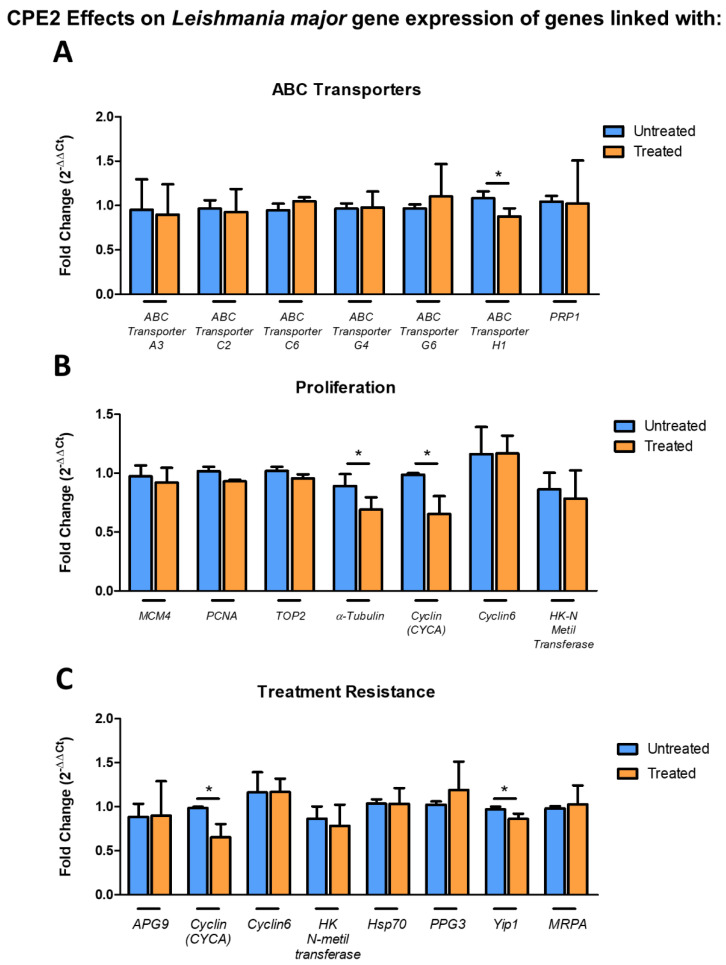
Effect of CPE2 on *L. major* promastigotes gene expression. mRNA levels of (**A**) selected ABC transporter genes, (**B**) genes related to proliferation, (**C**) genes associated with treatment resistance after 48 h of treatment with compound CPE2 (50 µM). The bars represent the means ± the SD from three independent experiments. (*, *p* < 0.05).

**Table 1 ijms-22-10493-t001:** Molecular dynamics stability results. Maximum and average α-Carbons RMSD values were recorded for each model and docked ligand and presented in the following table. Blank spaces mean no ligand–model interaction was predicted.

	Maximum RMSD (Å)			Average RMSD (Å)			Average Interacting Residues RMSD (Å)		
	Model			Model			Model		
Ligand	Mult1_lr	Mult1_lig	SwissModel_2	Mult1_lr	Mult1_lig	SwissModel_2	Mult1_lr	Mult1_lig	SwissModel_2
(None)	3.84	6.75	3.77	3.02	4.83	2.84			
CPE1	5.26	4.48		3.87	3.57		1.46	1.37	
CPE2	4.65			3.51			1.00		
CPE3	5.12			4.10			2.00		
CPE4	4.04			3.26			1.96		
CPE5		5.14			3.80			1.71	
CPE6		4.89			3.69			1.73	
CPE7			4.43			3.23			1.73

**Table 2 ijms-22-10493-t002:** Pharmacokinetics of selected and reference compounds. GI: Predicted gastrointestinal absorption. BBB: Predicted blood–brain barrier permeation. P-gp: Predicted substrate of P glycoprotein. CYP inhibitor: Likelihood of interactions with the five most common isoforms of cytochromes P450.

Molecule	GI Absorption	BBB Permeant	P-gp Substrate	CYP1A2 Inhibitor	CYP2C19 Inhibitor	CYP2C9 Inhibitor	CYP2D6 Inhibitor	CYP3A4 Inhibitor
CPE1	Low	No	No	No	No	Yes	No	No
CPE2	High	Yes	Yes	Yes	Yes	Yes	Yes	Yes
CPE3	High	No	Yes	No	No	No	No	No
CPE4	High	No	Yes	No	No	No	No	No
CPE5	High	No	Yes	No	No	No	Yes	Yes
CPE6	High	Yes	No	Yes	No	No	No	No
CPE7	High	Yes	No	No	Yes	No	Yes	No
Amphotericin B	Low	No	Yes	No	No	No	No	No

**Table 3 ijms-22-10493-t003:** Medicinal chemistry properties of selected and reference compounds. Bioavailability score: Probability of a compound to have bioavailability > 10% in a rat model. PAINS: Pan assay interference compounds (promiscuous compounds or potentially problematic fragments). Lead-likeness: Molecular suitability for optimization (small size and less hydrophobicity are favored). Synthetic accessibility: From 1 (very easy) to 10 (very difficult).

Molecule	Bioavailability Score	PAINS #Alerts	Lead-likeness #Violations	Synthetic Accessibility
CPE1	0.56	0	1	3.27
CPE2	0.55	0	1	3.09
CPE3	0.55	0	1	3.51
CPE4	0.55	0	0	3.47
CPE5	0.55	0	1	3.54
CPE6	0.55	0	0	2.41
CPE7	0.85	0	0	2.36
Amphotericin B	0.17	0	1	10

**Table 4 ijms-22-10493-t004:** Physicochemical properties. MW: Molecular weight (g/mol). #H-bond acceptors: Number of hydrogen bond acceptors. #H-bond donors: Number of hydrogen bond donors. Consensus log*P*: Average of logarithm of compound partition coefficient between n-octanol and water, as calculated by several methods. Lipinski: Number of Lipinski rule of five violations. TPSA: Topological polar surface area.

Molecule	Canonical SMILES	MW	#H-Bond Acceptors	#H-Bond Donors	Consensus log*P*	Lipinski #Violations	TPSA (Å²)
CPE1	OC(=O)c1sc(cc1C)S(=O)(=O)NCCc1nn(c2c1cccc2)C	379.45	6	2	2.31	0	137.91
CPE2	O=C1Nc2c(C1)cc(cc2)C(NC(=O)c1cc2c([nH]1)cc(cc2C)F)C	351.37	3	3	3.11	0	73.99
CPE3	CCc1cccc(c1)NC(=O)CCC(=O)N1CCC2(CC1)NCCNC2=O	372.46	4	3	1.18	0	90.54
CPE4	O=C(Nc1sc2c(c1C(=O)N)CCC2)NCCN1CCOCC1	338.43	4	3	1.19	0	124.93
CPE5	COCc1cc(=O)n(c(n1)N1CCCC1)CC(=O)Nc1c(C)cc(cc1C)C	384.47	4	1	2.44	0	76.46
CPE6	O=C(Nc1cc(Cl)ccc1N(C)C)CCc1ncn[nH]1	293.75	3	2	1.69	0	73.91
CPE7	COc1ccc2c(c1)CN(CC2)C(=O)Cc1cccc(c1)C(=O)O	325.36	4	1	2.45	0	66.84
Amphotericin B	OC1CCC(O)C(O)CC(O)CC2(O)CC(O)C(C(O2)CC(C=CC=CC=CC=CC=CC=CC=CC(C(C(C(OC(=O)CC(C1)O)C)C)O)C)OC1OC(C)C(C(C1O)N)O)C(=O)O	924.08	18	12	−0.39	3	319.61

**Table 5 ijms-22-10493-t005:** EC_50_ and SI (selectivity index) of selected compound and amphotericin B on murine bone marrow derived macrophages (Mφ) and *Leishmania* spp. promastigotes at 48 h post treatment.

	Mφ	*L. major*		*L. amazonensis*		*L. infantum*	
Compound	EC_50_ (μM)	EC_50_ (μM)	SI	EC_50_ (μM)	SI	EC_50_ (μM)	SI
CPE-2	>200	58.13 ± 1.72	>3.4	62.44 ± 2.12	>3.2	101.85 ± 0.35	>2
Amphotericin B	4.1 ± 0.7	0.1 ± 0.03	41	0.2 ± 0.06	20.5	1.4 ± 0.3	2.9

**Table 6 ijms-22-10493-t006:** Selected known ligands used for enrichment factor (EF) calculation during initial docking steps.

Code	Structure/Sequence	Canonical SMILES	MW (g/mol)	Name (Literature)	References
Gossypol	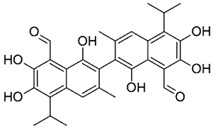	CC1=CC2=C(C(=C(C(=C2C(C)C)O)O)C=O)C(=C1C3=C(C4=C(C=C3C)C(=C(C(=C4C=O)O)O)C(C)C)O)O	518.6	Gossypol	PubChem CID: 3503 [15]
Peptide 1	**pS**PTF (**pS** = phosphoserine)		530.47	Peptide 1	[80]
Peptide M	D**F**DEYR**F**RKT (**F** = 4-fluoro-L-phenylalanine)		1412.47	Peptide 8.6	[26]

**Table 7 ijms-22-10493-t007:** Model evaluation indicated that Mult1_lr model performed the best at distinguishing decoys and known ligand compounds. Model name, used software, template, and quality parameters (QMEAN, Verify3d, ERRAT, and EF1) of the selected models were presented in the following table. Despite the performance of Mult1_lr, the best scoring model (best parameters in everything but EF1) was swissmodel_2.

Model Name	Used Method	Used Template (PDB Code)	QMEAN4	Verify3d	ERRAT	EF1	% of Aa in Ramachandran Favored Regions
SwissModel_2	SwissModel web server, YASARA force field	3PC6	0.71	84.95%	89.28	na	92.31%
Mult1_lr	MODELLER, using multiple alignment of protein sequences, de novo modeling of loops	2EP8 3U3Z 4BU0 2COU 4ID3 3OLC	0.64	68.24%	38.89	13.55	92.31%
Mult1_lig	MODELLER, multiple alignment of protein sequences, modeling with ligands in the binding site	Same as Mult1_lr +5U6K	0.33	68.75%	16.67	3.39	75.00%

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
