# Peer review of "Discovery and Validation of Lmj_04_BRCT Domain, a Novel Therapeutic Target: Identification of Candidate Drugs for Leishmaniasis"

_ijms, 2021, doi:10.3390/ijms221910493_

Round 1

Reviewer 1 Report

The authors studied the computer-generated models and identified candidate compounds targeting the BRCT protein binding interface.  The candidate compounds were tested in vitro, and a major hit was identified that inhibit the parasite growth in vitro.  This study provides a great example of in silico modeling and virtual screening can be used to guide drug discovery targeting the protein-protein interaction interface.  What’s vague to me from reading the manuscript is how and why the BRCT was selected to be analyzed.  The introduction of the manuscript (lines 104-149) can be restructured better to help the readers understand what a BRCT domain is, how it was chosen to be the study target. 

Section 2.8 Gene expression quantification: Line 112:  please change the “retrotranscription” to reverse transcription.  Gene expression study was not designed properly.  It should include a control (no treatment condition and you compare the gene expression of the target genes under CPE2 treatment and no treatment conditions.  The fold change should be calculated using the Pfaffl method or Livak method.

Section 3.1 Model generation and evaluation:  Line 143-146 should be removed. 

Author Response

Thank you.

Reviewer 2 Report

This study was performed to characterize Lmj_04_BRCT domain as novel therapeutic target in Leishmania Spp. Authors performed in silico and in vitro studies and identified CEP2 as a potential compound that may interfere the function of BRCT domain involved in multiple essential biological processes, including DNA damage repair, and cell-cycle control.

Overall, manuscript is written with sufficient background information. Below, a set of specific comments are listed to improve the manuscript.

  1. Authors need to summarize existing drugs in a supplementary table and reduce the text from the introduction.
  2. Authors need to summarize all the previous approaches in a supplementary table and reduce the text from the introduction.
  3. In line 32, the size of the filter disks should be 0.2 µm.
  4. In line 143, authors forgot to remove reviewer's comment from previous journal.
  5. Did author test hemolytic activity of the CPE2?
  6. Why 40 µM CPE2 showed higher activity than 100 µM CPE2?
  7. Did authors run the experiment longer period?
  8. Often successful drug works at sub-µM range, therefore author may need to find more potent derivative of CEP2.
  9. In figure 12, authors need to present the fold change in mRNA expression due to the exposure to CEP2 compared to control.
  10. In the discussion section, authors should not repeat the method and results. Authors should summarize the key finding of the study and discuss how this finding align or contradict with previous observations. Finally, authors need to describe what future studies are required to develop CEP2 as therapeutics.

Author Response

Thanks.

Round 2

Reviewer 1 Report

The revised version has addressed the comments and is good for publication.  Just a minor fix, Figure 11C is being cut off on the right.